Corrected: Publisher Correction

# Conversion of triphenylphosphine oxide to organophosphorus via selective cleavage of C-P, O-P, and C-H bonds with sodium

Jian-Qiu Zhang[1,2], Jingjing Ye[1,2], Tianzeng Huang[1,2], Hiroyuki Shinohara[3], Hiroyoshi Fujino[3] & Li-Biao Han [1,2]*

For over half a century, thousands of tons of triphenylphosphine oxide $Ph_3P(O)$ have been produced every year from the chemical industries as a useless chemical waste. Here we disclose efficient transformations of $Ph_3P(O)$ with cheap resource-abundant metallic sodium finely dispersed in paraffin oil. $Ph_3P(O)$ can be easily and selectively transformed to three reactive organophosphorus intermediates—sodium diphenylphosphinite, sodium 5H-benzo[b]phosphindol-5-olate and sodium benzo[b]phosphindol-5-ide—that efficiently give the corresponding functional organophosphorus compounds in good yields. These functional organophosphorus compounds are difficult to prepare but highly industrially useful compounds. This may allow $Ph_3P(O)$ to be used as a precious starting material for highly valuable phosphorus compounds.

[1] National Institute of Advanced Industrial Science and Technology (AIST), Tsukuba, Ibaraki 305-8565, Japan. [2] Division of Chemistry, Faculty of Pure and Applied Sciences, University of Tsukuba, Tsukuba, Ibaraki 305-8571, Japan. [3] Katayama Chemical Industries Co., Ltd., 26-22, 3-Chome, Higasinaniwa-cho, Amagasaki, Hyogo 660-0892, Japan. *email: libiao-han@aist.go.jp

Triphenylphosphine oxide $Ph_3P(O)$ is a chemically stable compound that is primarily generated, tens of thousands tons a year, as a by-product from the chemical industries, during the preparation of valuable fine chemicals, such as vitamins, pharmaceuticals, agrochemicals etc, and bulk chemicals, such as butanols, using triphenylphosphine $PPh_3$ as an oxophilic reagent or ligand for a metal catalyst (Supplementary Note 1)[1–8]. A well-known serious problem associated with $Ph_3P(O)$ is that thousands of tons of this compound are discarded as useless chemical waste because of its limited utilities. This situation has lasted for more than half a century, and has become a big concern from both industry and academia sides[9,10]. In order to solve this problem, extensive studies have been carried out world widely. Among them, the reduction of triphenylphosphine oxide $Ph_3P(O)$ to its original form triphenylphosphine $Ph_3P$ is most studied[11]. However, either hard conditions or expensive reductants are required in order to break the strong P=O bond. Therefore, a practically operable way that can settle the triphenylphosphine oxide problem has not been found yet[12–15].

Herein we disclose a potentially effective solution to this longstanding problem (Fig. 1). By treatment with the cheap, resource-abundant metallic sodium (sodium finely dispersed in paraffin oil with μm-scale sizes; hereafter abbreviated as SD) at 25 °C, triphenylphosphine oxide $Ph_3P(O)$, the so far discarded chemical waste (A), can be transformed, easily and selectively, to a variety of organic phosphorus compounds (phosphoryl compounds and phosphines) that are widely used valuable chemicals in the industry[16]. It is noted that the current process for the production of these organophosphorus compounds is rather dirty, energy-consuming and dangerous since it starts from the highly toxic benzene and phosphorus trichloride (C)[17,18]. Heavy pollution problems also associate with their preparation because of the poor efficiency. Therefore, the present new finding not only provides a use for waste $Ph_3P(O)$ (**1**) but also may aid the preparation of other useful organophosphorus compounds (**2**).

As depicted in Fig. 1d, by cleaving one C-P bond, sodium diphenylphosphinite **2** is generated quantitatively. This intermediate **2** is easily transformed to the corresponding phosphine oxides **2′** (a). On the other hand, by slightly changing the conditions, sodium 5H-benzo[b]phosphindol-5-olate **3** is generated in high yields via one C-P and two C-H bonds cleavage (b). More interestingly, the O-P bond in **3** can be further cleft to sodium benzo[b]phosphindol-5-ide **4** quantitatively (c). Thus, by simply treating with metallic sodium, triphenylphosphine oxide can readily produce three kinds of highly valuable phosphorus compounds.

## Results

**Reactions of $Ph_3P(O)$ with metallic sodium.** We serendipitously discovered this rapid reaction of $Ph_3P(O)$ with metallic sodium as we added $Ph_3P(O)$ to THF that contains trace amount of metallic sodium initially used for its drying. The originally transparent colorless THF solution of $Ph_3P(O)$ (Fig. 2a), instantly changed to yellow and then brown (Fig. 2b) at 25 °C. This clear color change is a strong indication that a rapid chemical reaction takes place between $Ph_3P(O)$ and sodium. Indeed, this is true. A subsequent experiment surprisingly revealed that, at 25 °C, upon adding sodium to $Ph_3P(O)$ dissolved in THF, an exothermic reaction took place rapidly and the starting material $Ph_3P(O)$ was completely consumed after 2 h (Fig. 3, equation (1)).

This easy conversion of $Ph_3P(O)$ by sodium was rather unexpected considering that until now literatures all had to conduct this reaction in a highly reducing medium by dissolving sodium in liquid ammonia (the Birch reduction medium) at low temperatures[12,13]. In addition to the tedious process under the Birch reduction system, that requires difficult handling and operating techniques, yields and selectivity of the desired products were also not satisfactory. A NaH/LiI composite system was recently reported to break down the C-P bond of triarylphosphine oxides. However, NaH is a rather costly reagent. Moreover, a long-time heating (overnight at 60 °C) was required[15].

As shown in Fig. 3, metallic sodium (2.5 mmol), cut to small pieces, was added to $Ph_3P(O)$ (0.5 mmol) dissolved in THF (3 mL) at 25 °C (equation (1)). The color of the reaction mixture soon changed to brown. After stirring for 2 h, $^{31}P$ NMR spectroscopy showed that the starting material $Ph_3P(O)$ at 32.9 ppm almost disappeared, and three new signals emerged at 91.1 ppm (compound **2**) and 102.1 ppm (compound **3**) and 6.7 ppm (compound **4**) after overnight stirring. In order to identify these compounds, *n*-BuBr was added to the solution and **2′**, **3′**, and **4′** were obtained in 70%, 8%, and 21%, respectively, confirming that the new generated phosphorus species are sodium

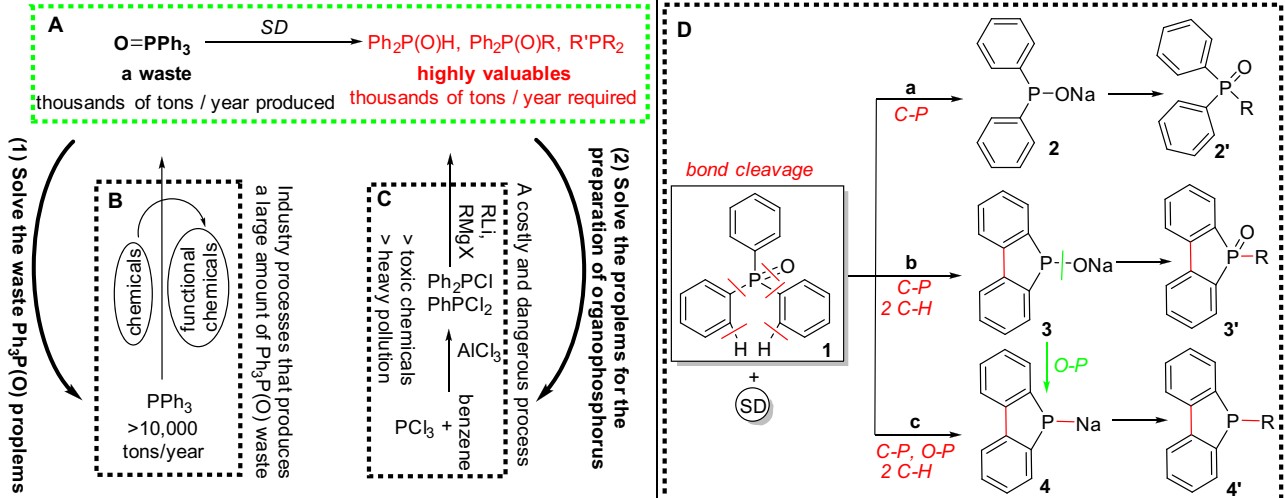

**Fig. 1 Conversion of waste $Ph_3P(O)$ to valuable organic phosphorus compounds. a** Transformation of $Ph_3P(O)$ to a variety of organic phosphorus compounds by SD. **b** A vast amount of $Ph_3P(O)$ is produced and discarded as a useless chemical waste. **c** Costly and dangerous methods for the preparation of organophosphorus compounds. **d** Selective C-P, C-H and O-P bond cleavage of triphenylphosphine oxide by SD under mild conditions. Condition a: **1** and SD in THF at 25 °C; Condition b: SD/PhCl then **1** at 25 °C, THF; Condition c: SD/PhCl, **1**, then SD at 25 °C, THF.

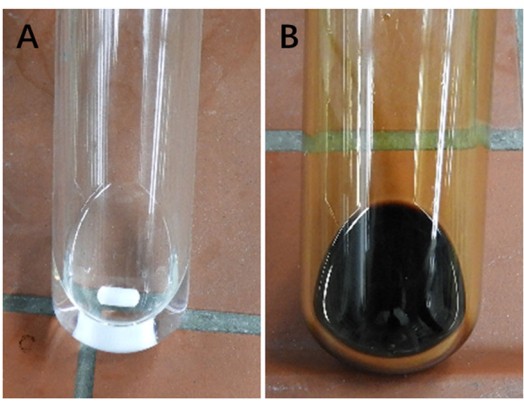

**Fig. 2 Reactions of Ph₃P(O) with metallic sodium. a** Ph₃P(O) dissolved in THF. **b** Ph₃P(O) dissolved in THF in the presence of Na.

diphenylphosphinite (**2**), sodium 5H-benzo[b]phosphindol-5-olate (**3**) and sodium benzo[b]phosphindol-5-ide (**4**) (Supplementary Figs. 9–11).

**Selective generation of 2 by transformation of Ph₃P(O) to Ph₂P(O)H and Ph₂P(O)R.** More excitingly, in addition to the disclosure of the event that Ph₃P(O) could readily react with sodium under mild conditions, the reaction conditions for the generation of the three phosphorus species **2**, **3**, and **4** were tunable, so that these three active phosphorus species could be highly selectively formed, respectively. First, instead of sodium lump, when sodium powder dispersed in paraffin oil (average particle size < 10 μm, here below abbreviated as SD)[19] was used, we can produce compound **2** exclusively within a few minutes (ref. [19] Being similar to Na, other alkali metals are expected to react with Ph₃P(O) too. Indeed, the reaction of Ph₃P(O) with lithium dispersion (metallic lithium finely dispersed in paraffin

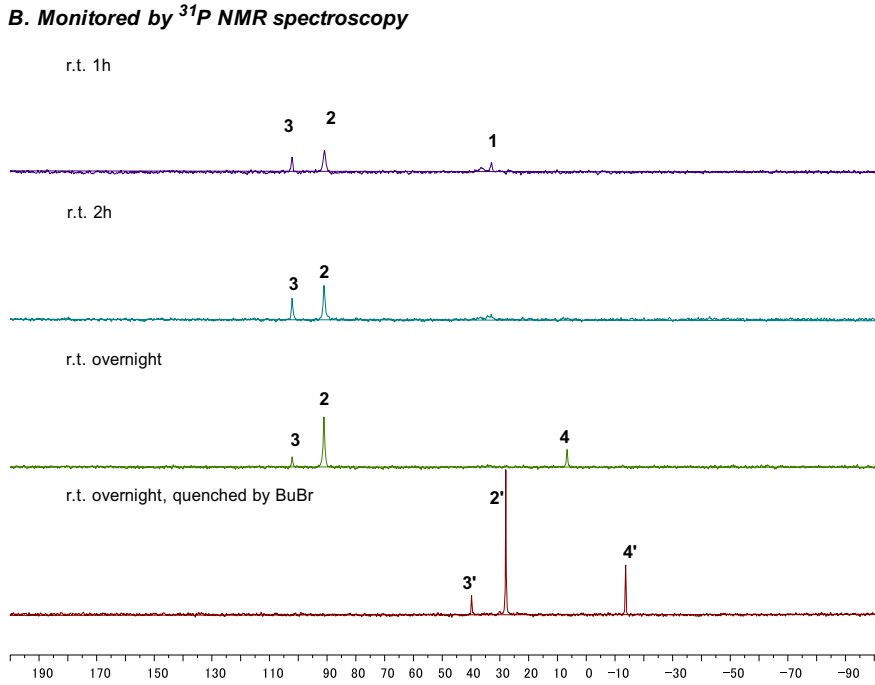

**Fig. 3 Reaction of triphenylphosphine oxide with sodium. a** Triphenylphosphine oxide **1** rapidly reacted with sodium at 25 °C. Reaction conditions: 0.5 mmol Ph₃P(O) was dissolved in 3 mL THF, and 2.5 mmol metallic Na was added at 25 °C. **b** The reaction mixture was stirred for 1 h, 2 h and overnight and monitored by ³¹P NMR, respectively.

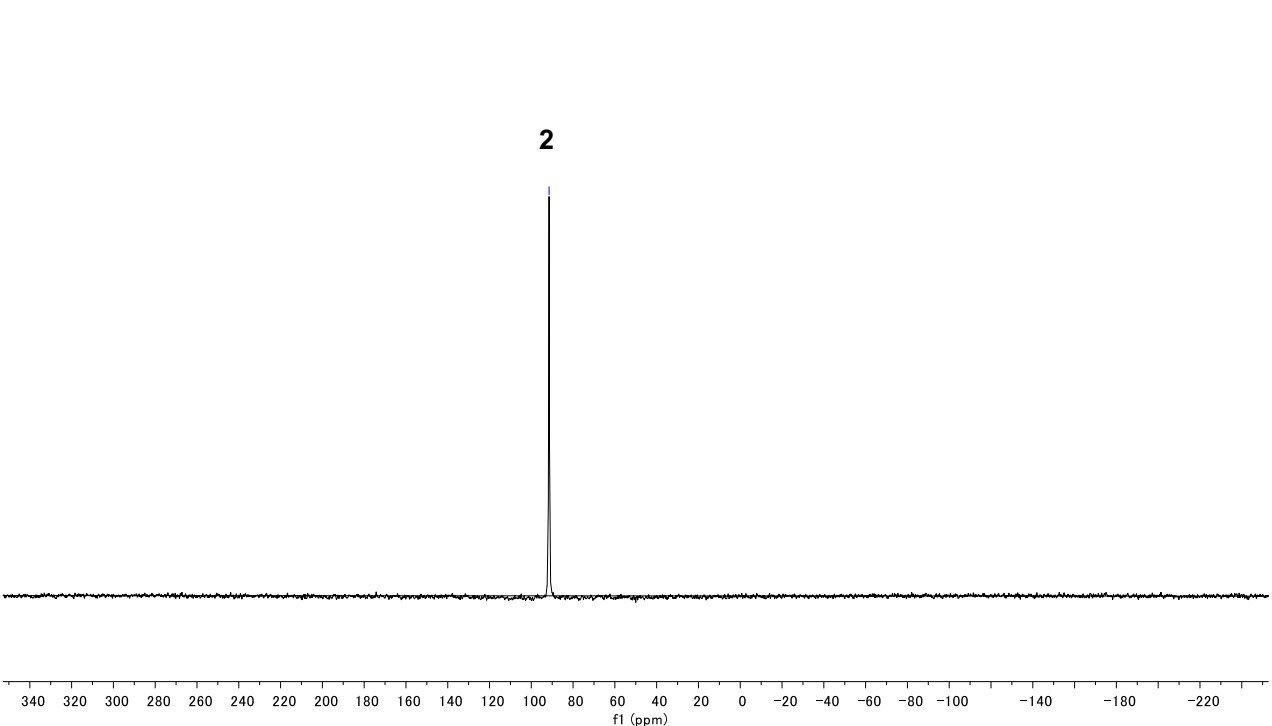

**Fig. 4 Synthesis of 2. a** Selective transformation to sodium diphenylphosphinite **2**. Reaction conditions: sodium (SD) (2.5 mmol) was added to Ph₃P(O) (1.0 mmol) dissolved in 5 mL THF at 25 °C and the mixture was stirred for 10 min. **b** NMR spectra copy of the crude reaction solution.

oil) also proceeded rapidly at 25 °C. However, being different to the reaction with Na, a lot of side products were obtained with Li and the selectivity to Ph₂POLi was only ca. 70%.). Thus, 2.5 mmol SD was added to 1.0 mmol Ph₃P(O) in 5 mL THF at 25 °C (Fig. 4, equation (2)). After 10 min, the corresponding sodium diphenylphosphinite (**2**, $\delta = 91.5$ ppm) was produced exclusively and quantitatively! No side products could be detected at all!

As to the molar ratios of sodium vs Ph₃P(O), we found that more than 2 equivalents of sodium are necessary for the selective complete conversion of Ph₃P(O) to Ph₂P(ONa). For example, under similar conditions, when one equivalent SD was used, 12% Ph₃P(O) remained unchanged, and Ph₂P(ONa) and **3** were obtained in 60% and 28% yield, respectively, as determined by ³¹P

NMR spectroscopy. Therefore, the reaction should proceed, being similar to that under the super reducing medium Na/NH₃[12,13], via the cleavage of one C-P bond of Ph₃P(O) by two equivalents of sodium generating an equimolar Ph₂P(ONa) and PhNa (vide infra) (Fig. 4).

This easy and quantitative conversion of **1** Ph₃P(O) to **2** Ph₂PONa guaranteed its application as an industrially useful reaction because, now, diphenylphosphine oxide Ph₂P(O)H, a widely used but rather expensive industrial chemical, can be easily prepared from the waste chemical Ph₃P(O)! Diphenylphosphine oxide Ph₂P(O)H[20] is widely employed as a versatile starting material for the synthesis of a lot of valuable organophosphorus compounds. This compound is currently industrially produced

via the hydrolysis of Ph$_2$PCl. However, Ph$_2$PCl is prepared from a rather inefficient Friedel-Crafts reaction of PCl$_3$ and benzene using AlCl$_3$ that releases a large amount of wastes (more than 3 tones wastes in order to produce one tone product)[17,18]. As shown in Fig. 5, by simply adding water, the intermediate sodium phosphinite **2** developed above could quantitatively give Ph$_2$P(O)H **2–1** (Fig. 5a). It is noted that Ph$_2$P(O)H is also a starting material for the preparation of 2,4,6-trimethylbenzoyldipenylphosphine oxide (TPO). TPO is a important photoinitiator and thousands of tons of TPO are broadly used in the realm of photopolymerization[21]. This compound is industrially prepared by two methods: (1) Michaelis–Arbuzov reaction of alkoxyphosphine with acyl chloride[22,23] and (2) oxidation of α-hydroxyphosphine oxide generated by the addition of Ph$_2$P(O)H to the aldehyde[24] (Fig. 5b). We found that TPO could be conveniently generated directly using Ph$_2$PONa **2** that can eliminate the isolation of Ph$_2$P(O)H and other steps for the synthesis of TPO (Fig. 5b). For

example, by adding Ph$_2$PONa **2** to 2,4,6-trimethylbenzoyl chloride (MesC(O)Cl) in THF at 0 °C, the desired product TPO **2–2** was obtained in 56% yield. Beyond its practical utility, it is noted that this reaction is the first example for the preparation of TPO and analogues by the direct nucleophilic substitution reactions with an acylchloride because all of the literature attempts generated side products rather than the desired product TPO[21–24].

Finally, Ph$_2$PONa can also efficiently react with an organohalide to give the corresponding phosphine oxide in high yield which is useful in organic synthesis, metal extraction etc. (Fig. 5c)[25,26]. Although the nucleophilic substitution reaction of Ph$_2$PONa with RBr generating Ph$_2$P(O)R is a known reaction, the high yield of Ph$_2$P(O)R with a slightly excess RBr is surprising considering that an equimolar PhNa is also generated in the reaction mixture but does not interfere in the nucleophilic substitution reaction (vide infra).

**Fig. 5 Utility of sodium diphenylphosphinite 2.** Reaction conditions: Ph$_2$PONa **2** was prepared from 1.0 mmol Ph$_3$P(O) in 5 mL solvent and 2.5 mmol SD according to the standard procedure. **a** 2.0 mL saturated aqueous NH$_4$Cl solution was added to Ph$_2$PONa **2** (1.0 mmol, in 1,4-dioxane) at 25 °C. **b** Ph$_2$PONa **2** (1.0 mmol, in THF) was added to MesC(O)Cl (1.5 mmol) at 0 °C. **c** An alkyl halide (1.2 mmol) was added to Ph$_2$PONa **2** (1.0 mmol, in THF) at 0 °C. Isolated yield.

**Fig. 6 Possible mechanisms. a** The mechanism to **3** using PhNa. **b** The mechanism to **6** using PhLi (ref. [28]).

**Selective generation of 3**. Fixing the optimized conditions for the selective generation of **3** was not as easy as **2**. The study hardly progressed until we eventually realized that PhNa should be the key for its generation (Fig. 6). Thus, we anticipated that sodium 5H-benzo[b]phosphindol-5-olate **3**, would be formally generated by dehydrogenative cyclization. Since PhNa **5** was generated during the reaction of Ph₃P(O) with Na, PhNa might act as a base to react with Ph₃P(O) to give **3** via cyclization[27,28]. An early literature reported that the reaction of PhLi with Ph₃P(O) in THF under reflux overnight gave 5-phenyl-5H-benzo[b]phosphindole **6** rather than **3** (Fig. 6b)[27]. Although not fully understood at present, this difference in reactivity between PhLi and PhNa is very interesting. It should be noted that while the current reaction with PhNa took place rapidly at room temperature, the reaction with PhLi required a long-time heating[27].

This was indeed the case. When Ph₃P(O) (0.45 mmol dissolved in 2 mL THF) was added to PhNa (1.0 mmol prepared from 2.0 mmol SD with 1.1 mmol PhCl) at 25 °C (Table 1, equation (3))[20], 5H-benzo[b]phosphindol-5-olate **3** ($\delta = 101.7$ ppm) was generated predominantly (Table 1, run 1). By quenching the organophosphorus species with *n*-OctBr, **3′b** was obtained in

77% yield together with **2–3b** generated via the reaction of **2** (11%), respectively, as determined by ³¹P NMR spectroscopy. Efforts had been devoted to improving the yield and selectivity of **3**. Switching the ratio of Ph₃P(O) **1** and PhNa to 0.33:1 leaded to lower yield and selectivity of **3** (Table 1, run 2). When a solid PhNa was added to Ph₃P(O) dissolved in THF, **3** and **2** were generated in 42% and 38% respectively (Table 1, run 3). Interestingly, by increasing the amount of Ph₃P(O) to 0.9 mmol, nearly an equimolar ratio to PhNa, a similar yield of **3** could be obtained (Supplementary Fig. 5), indicating that only one equivalent of PhNa is required for the generation of **3** in the reaction, which not only significantly improved the efficiency of the use of PhNa (Table 1, run 4), but also sustained the proposed mechanism (Fig. 6). The selectivity was not further improved either under a lower or higher temperature (Table 1, runs 5–6).

Molecules with dibenzophosphole framework have great potentials as novel optical and electrical materials[29]. As shown in Fig. 7, old procedures for the synthesis of dibenzophosphole oxides mainly relied on metathesis reaction between dilithiated biphenyl with RPCl₂ followed by oxidation (a)[30]. Recently, a palladium-catalyzed intramolecular arylation of

---

**Table 1 Selective transformation of Ph₃P(O) to sodium 5H-benzo[b]phosphindol-5-olate 3.**

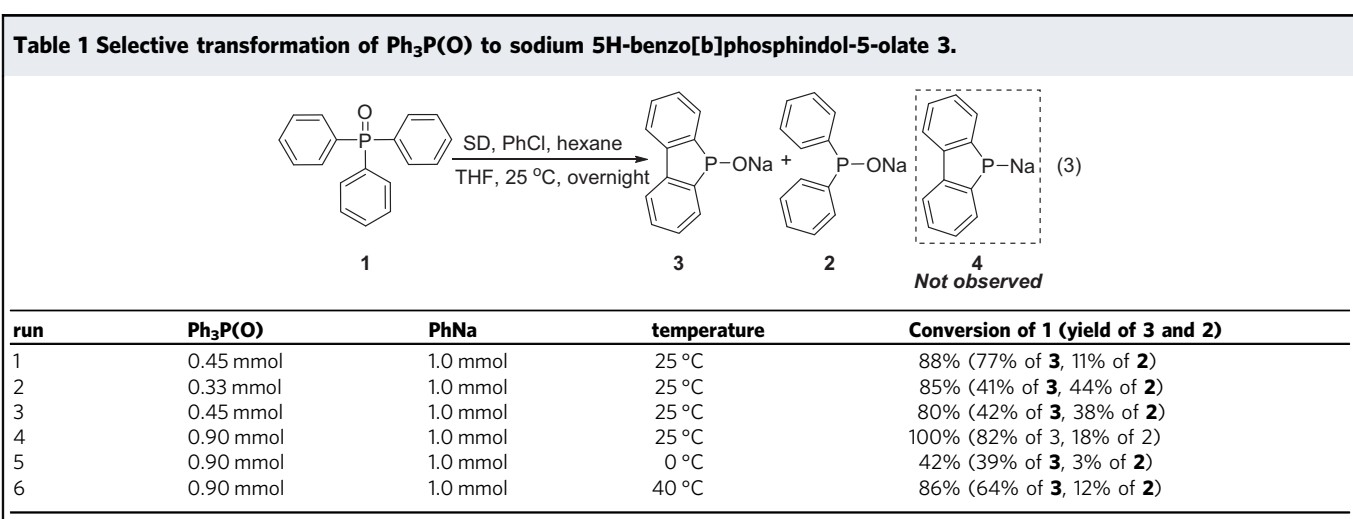

| run | Ph₃P(O) | PhNa | temperature | Conversion of 1 (yield of 3 and 2) |
|---|---|---|---|---|
| 1 | 0.45 mmol | 1.0 mmol | 25 °C | 88% (77% of **3**, 11% of **2**) |
| 2 | 0.33 mmol | 1.0 mmol | 25 °C | 85% (41% of **3**, 44% of **2**) |
| 3 | 0.45 mmol | 1.0 mmol | 25 °C | 80% (42% of **3**, 38% of **2**) |
| 4 | 0.90 mmol | 1.0 mmol | 25 °C | 100% (82% of 3, 18% of 2) |
| 5 | 0.90 mmol | 1.0 mmol | 0 °C | 42% (39% of **3**, 3% of **2**) |
| 6 | 0.90 mmol | 1.0 mmol | 40 °C | 86% (64% of **3**, 12% of **2**) |

Reaction conditions: PhNa was generated in situ by the reaction of SD (2.0 mmol) with PhCl (1.1 mmol) in 2.0 mL hexane at 25 °C for 1 h. A specified amount of Ph₃P(O) dissolved in 2.0 mL THF was then added into PhNa at 25 °C and the mixture was stirred for overnight. Yields were estimated from ³¹P NMR spectroscopy based on **1** used. ᵃSolid PhNa was used

---

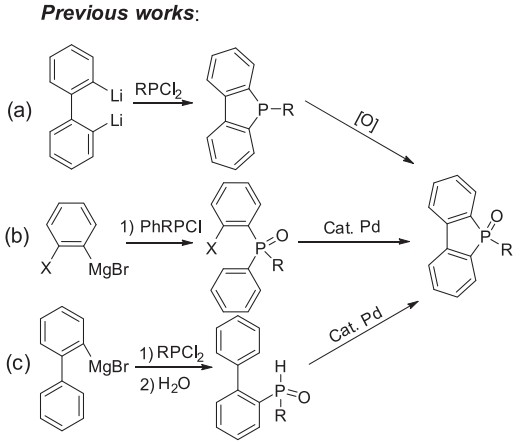

**Fig. 7 Efficient synthesis of dibenzophosphole oxides from the chemical waste Ph₃P(O).** Reaction conditions: PhNa was generated in situ by adding PhCl (1.1 mmol) to a suspension of SD (2.0 mmol in 2.0 mL hexane) at 25 °C for 1 h. Ph₃P(O) dissolved in THF (2.0 mL) was added into the above PhNa suspension and stirred for overnight. RBr (1.5 mmol) was then added at 0 °C and stirred for 0.5 h. Isolated yield.

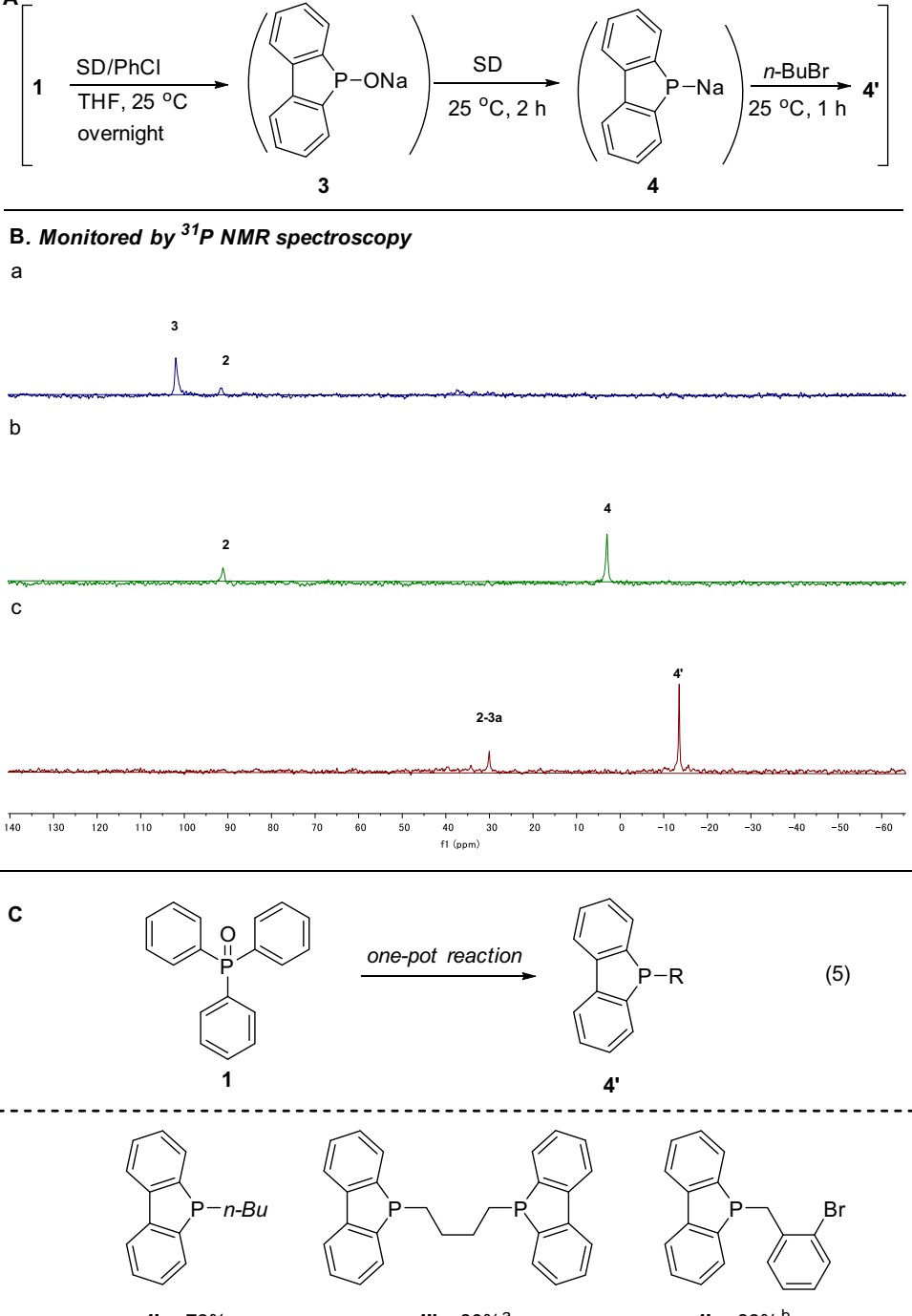

**Fig. 8 a** Selective transformation of $Ph_3P(O)$ to sodium benzo[b]phosphindol-5-ide **4**. **b** Reaction conditions and monitored by [31]P NMR: (a) sodium H-benzo[b]phosphindol-5-olate **3** was generated in situ from $Ph_3P(O)$ (0.9 mmol) and PhNa (1.0 mmol); (b) SD (2.5 mmol) was then added and stirred for 2 h; (c) n-BuBr (2.0 mmol) was added to the mixture and stirred for 1 h. **c** One-pot conversion of $Ph_3P(O)$ to phosphle **4′**. Isolated yield. [a]0.4 mmol of $BrC_4H_8Br$ was added. [b]The product was oxidized by $H_2O_2$ for easy isolation.

ortho-halodiphenylphosphine (b) and intramolecular dehydrogenative cyclization of secondary hydrophosphine oxides (c) have been developed[31,32]. All these approaches have to use the toxic phosphine chlorides, and the processes are tedious. As shown in Fig. 7d, the sodium 5H-benzo[b]phosphindol-5-olate **3** derived from $Ph_3P(O)$ was easily transformed to the corresponding dibenzophosphole oxides in moderate to high yields. Functional groups like Cl, CN, $CF_3$ are well tolerable. Therefore, an efficient

way for the generation of these useful dibenzophosphole oxides by using the chemical waste $Ph_3P(O)$ was established.

**Selective generation of 4 through P=O reduction by sodium.** A more fascinating phenomenon is that even the trivalent phosphole **4** can be selectively generated starting from $Ph_3P(O)$ (Fig. 8). Thus, during the study on further possible reactions with metallic sodium of **2** and **3**, we surprisingly found that although no

**Fig. 9 10-g scale reaction. 1** was treated with SD in dioxane followed by quenching with water to give **2–1** in a high yield.

reaction took place with **2**, **3** reacted quickly to give **4**! Thus, [31]P NMR showed that after the addition of SD to **3** at 25 °C for a few minutes, the signal of **3** completely disappeared and a new signal of **4** at 3.0 ppm appeared. As expected, the subsequent addition of *n*-BuBr to the mixture gave compound **4′** ($\delta = -13.5$ ppm) which was fully characterized by comparing with an authentic sample prepared separately[33]. This protocol is amenable to use dibromides as electrophiles for the synthesis of bisphosphole **4′b**. Since the reaction of sodium benzo[b]phosphindol-5-ide **4** with an alkyl bromide is faster than that with an aromatic bromide, **4′c** could be selectively generated. Therefore, by carrying out a one-pot reaction, the so far chemical waste Ph₃P(O) could also be easily converted to phosphole **4′** (Fig. 8, equation (5))! This is also a rare example for converting phosphine oxide (P(V)) to phoshine (P(III)) that usually requires highly reactive reducing reagents such as LiAlH₄ and hydrosilanes, or under harsh conditions, in order to break down the robust P-O bond[11–15].

**A 10-gram scale reaction**. As demonstrated by a lab-scale reaction, the present method is easily applicable to a large-scale preparation of the phosphorus compound (Fig. 9). For example, after treating 10 g of Ph₃P(O) **1** with 90 mmol SD (2.5 equiv.) at 25 °C, water was added. The mixture was simply extracted with EtOAc, washed by hexane and passed through a short silica gel column. A spectroscopically pure diphenylphosphine oxide **2–1** was obtained as a white solid (6.93 g, 96% yield).

In conclusion, we disclosed that the C-P bond of Ph₃P(O) can be efficiently cleft by metallic sodium under normal conditions without using dangerous super reducing media (sodium dissolved in ammonia Na/NH₃). Three basic reactive organophosphorus intermediates **2**, **3**, **4** can be selectively generated from the combination of Ph₃P(O) with metallic sodium, giving the corresponding organophosphorus compounds efficiently (Fig. 1). The industrial markets for these organophosphorus compounds derivatives, that are difficult to prepare by other methods, is large enough to consume up all of the Ph₃P(O) produced as a chemical waste from the chemical industry. Therefore, Ph₃P(O) may serve as a precious chemical stock for highly valuable functional organophosphorus compounds. We believe that this finding can settle the Ph₃P(O) problem that has annoyed people for half a century.

## Methods

**Synthesis of compound 2**. Under argon, SD (0.25 mL, 2.5 mmol) was added to Ph₃P(O) (278 mg, 1.0 mmol) dissolved in THF (5 mL) at 25 °C with stirring. The initially colourless transparent solution turned to brown soon. After stirring for 10 min, compound **2** was obtained quantitatively. By adding *n*-BuBr (130 µL, 1.2 mmol) to the above mixture at 0 °C, **2–3a** was obtained.

**Synthesis of compound 3**. PhCl (112 µL, 1.1 mmol) was added dropwise to SD (0.2 mL, 2.0 mmol) suspended in hexane (2.0 mL) at 25 °C under argon. After stirring for 1 h, Ph₃P(O) (250 mg, 0.9 mmol) dissolved in THF (2.0 mL) was added and the mixture was stirred at 25 °C overnight to give compound **3**. By quenching the reaction mixture thus generated with *n*-OctBr (258 µL, 1.5 mmol) at 0 °C afforded **3′b**.

**Synthesis of compound 4**. Under nitrogen, to a solution of compound **3** obtained above was added SD (0.25 mL, 2.5 mmol) at 25 °C. The mixture was stirred for 2 h giving compound **4**. By quenching with *n*-BuBr (215 µL, 2.0 mmol) at 0 °C, **4′** was obtained.

**Product derivatizations**. Full procedures for transformations of **1** to **2**, **3**, **4** and their derivatives **2′**, **3′**, **4′** are available in the Supplementary Methods and Supplementary Figs. 1–6.

**NMR spectra**. [1]H,[13]C and [31]P NMR Spectra of all products were provided. See Supplementary Figs. 12–50.

## Data availability
The authors declare that all the data supporting the findings of this study are available within the paper and its supplementary information files, and also are available from the corresponding author upon reasonable request.

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

## Acknowledgements

This study was partially supported by a joint-research of National Institute of Advanced Industrial Science and Technology (AIST) with Katayama Chemical Industries Co., Ltd. Yuka Mino and Eiichi Ikawa were acknowledged for their participation. We thank Kobelco Eco-Solutions Co., Ltd. for providing SD samples.

## Author contributions

L-B.H. discovered the rapid color change as Na was added to a THF solution of **1** and identified **2**. J.Y. isolated **4′a** and **4′b**. T.H. isolated **2–3a** and **2–3b**, H.S. and H.F. prepared **3′a** and **3′b**. J-Q.Z. conducted all the other experiments and prepared the Supplementary Information. L-B.H. oversaw the study and prepared the manuscript with J-Q.Z.

## Competing interests

The authors declare no competing interests.
