## [Peer Review File · Communications Chemistry]

Reviewers' comments:

Reviewer #1 (Remarks to the Author):

This manuscript by Han and co-workers describes a new use of sodium dispersion (SD) for controlled reductive transformation of triphenylphosphine oxide to diphenylphosphine oxide, dibenzophosphole oxides, and dibenzophosphole. Conversion of triphenylphosphine oxide to diphenylphosphine oxide can be implemented under Birch-type reaction conditions, which is well documented in the precedents. Thus, use of SD instead of dissolving metals should be more useful and practical, while being somewhat incremental in character. The reaction conditions enabling selective synthesis of dibenzophosphole oxides and dibenzophosphole would be rather appealing, although this work did not investigate much on the nature of the process such as substrate scope and substituent compatibility. I am wondering if phenyllithium has a similar reactivity for the formation of dibenzophosphole oxides, given the fact that generation of PhNa needs longer reaction time from SD and PhCl.

I am not very supportive to the way of writing of this manuscript, which sounds somewhat exaggerating. I would urge the authors to focus much more on scientific significance.

I do not recommend this work to be published as a current format.

Reviewer #2 (Remarks to the Author):

I read with an interest the manuscript "No Longer a Waste: Ready Conversion of Ph₃P(O) to Difficultly Accessible Organophosphorus via Unprecedented Selective Cleavage of C-P, O-P and C-H Bonds with Novel SD Na". The research by Professor Li-Biao Han and coworkers is promising for chemical industry. The solution of the recycling problem of Ph₃P(O) waste was offered and justified. The cheap resource-abundant metallic sodium fine dispersed in paraffin oil (SD Na) served a good alternative to more popular but expensive silanes. Ph₃P(O) waste was transformed into valuable chemicals in the industry. The applied synthetic approaches were mild, the experiments were conducted at 25 °C or lower, without complex catalysts and a hard work up. The authors described the advantages of their method over the current industrial process. They proposed the mechanisms of the reactions and gave the confirmation. I believe that the manuscript should be accepted after minor revisions.

1) I would recommend you to estimate influence of particle size of sodium. Try to conduct the experiment with little pieces of sodium instead of dispersed sodium. Even if the reaction is not working it would help a lot for other chemists to save time to evaluate this possibility.

2) In SI, please, point out the conditions for HPLC isolations. There is no data about solvent, flow rate, retention times, etc.

3) In SI, please, point out the solvent or system of solvents and ratio for every product which you isolated using recrystallization.

- 4) Please recheck manuscript and SI for the typos. E.g. in SI you should write Na₂SO₄ instead of NaSO₄.
- 5) In SI p. S2 (2.1 General method A) I would recommend you to write solvent instead of particular THF or 1,4-dioxane in general procedure (p. S2).
- 6) In SI p. S3 in (1) For the synthesis of Ph₂P(O)H you should clarify and write, for example, According to General method A, sodium diphenylphosphinite 2 was obtained in 1,4-dioxane and then to describe preparation of Ph₂P(O)H.
- 7) In SI p. S3 synthesis of Ph₂P(O)C(O)Mes. It is important product for industry which you isolate using HPLC. Please, try to isolate it using recrystallization if it is possible.
- 8) In General information you wrote that scale up synthesis of products were purified by column chromatography, but in the protocol (p. S6) you wrote that recrystallization was used to purify the product. What method of purification was really used?
- 9) Please, measure melting point for all solid compounds and compare it with the literature data.
- 10) In SI p. S12, Product 2-1, what signals are in the upfield ¹H NMR?
- 11) In SI you wrote "This compound is known" and give the reference. I would recommend you to write that your data "are in agreement with the literature data", because the fact that the substance is known doesn't show that your data correspond to the literature.
- 12) Please, provide the data about multiplicity and spin-spin coupling constants in ¹H NMR, although in ¹H NMR of 3'a (p. S18) and 3'b (p. S19), for example, I can see triplets in downfield. 3'a 7.56 (t, 2H, J = ?), 7.42 (t, 2H, J = ?). 3'b 7.5 (t, 2H, J = ?), 7.34 (t, 2H, J = ?). ¹H NMR of 3'a and 3'b the reference signal is 7.256 and 7.202 respectively. Please, correct the reference and other signals in these spectra.

Reviewer #3 (Remarks to the Author)

The revised manuscript is devoted to the well-known problem of how to recycle triphenylphosphine oxide back into organic synthesis. This compound is a waste byproduct of many transformations including triphenylphosphine. The importance of the latter is overwhelming as it is widely used both in organic synthesis (production of fine chemicals) and transition metal catalysis (the synthesis of otherwise difficultly accessible organic molecules). Presented attempts in recycling triphenylphosphine oxide are of highly practical value as it gives an access to very valuable reactants in a very facile and efficient way using cheap and readily accessible reagent. Regarding this I would recommend the editorial office to accept the manuscript for publication after minor style and language improvement.

Responds to reviewers' comments:

Reviewer #1 (Remarks to the Author):

This manuscript by Han and co-workers describes a new use of sodium dispersion (SD) for controlled reductive transformation of triphenylphosphine oxide to diphenylphosphine oxide, dibenzophosphole oxides, and dibenzophosphole. Conversion of triphenylphosphine oxide to diphenylphosphine oxide can be implemented under Birch-type reaction conditions, which is well documented in the precedents. Thus, use of SD instead of dissolving metals should be more useful and practical, while being somewhat incremental in character.

1) The reaction conditions enabling selective synthesis of dibenzophosphole oxides and dibenzophosphole would be rather appealing, although this work did not investigate much on the nature of the process such as substrate scope and substituent compatibility.

Response: Thanks very much for the valuable suggestion. As for the substrate scope and substituent compatibility, we added some substrates as shown in Fig. R1, due to the difficulty of purifying **3'g** and **3'h**, only ³¹P NMR yield were given.

Dibenzophosphole oxides:

Dibenzophospholes:

Fig. R1 Complementary substrates of dibenzophosphole oxides and dibenzophospholes

2) I am wondering if phenyllithium has a similar reactivity for the formation of dibenzophosphole oxides, given the fact that generation of PhNa needs longer reaction time from SD and PhCl.

Response: As shown in Fig. R2, PhLi is able to react with Ph₃P(O). This was a known reaction (Ogawa, S., Tajiri, Y., & Furukawa, N. *Bull. Chem. Soc. Jpn.* **64**, 3182–3184 (1991). However, the products were different. For PhLi, 5-phenyl-5H-benzo[b]phosphindole was obtained, rather than **3** for PhNa.

Fig. R2 The reaction of $\text{Ph}_3\text{P}(\text{O})$ with PhLi (Ogawa et al *Bull. Chem. Soc. Jpn.*)

3) I am not very supportive to the way of writing of this manuscript, which sounds somewhat exaggerating. I would urge the authors to focus much more on scientific significance.

Response: he is right. We've toned down accordingly.....

Reviewer #2 (Remarks to the Author):

I read with an interest the manuscript No Longer a Waste: Ready Conversion of $\text{Ph}_3\text{P}(\text{O})$ to Difficultly Accessible Organophosphorus via Unprecedented Selective Cleavage of C-P, O-P and C-H Bonds with Novel SD Na". The research by Professor Li-Biao Han and coworkers is promising for chemical industry. The solution of the recycling problem of $\text{Ph}_3\text{P}(\text{O})$ waste was offered and justified. The cheap resource-abundant metallic sodium fine dispersed in paraffin oil (SD Na) served a good alternative to more popular but expensive silanes. $\text{Ph}_3\text{P}(\text{O})$ waste was transformed into valuable chemicals in the industry. The applied synthetic approaches were mild, the experiments were conducted at 25 °C or lower, without complex catalysts and a hard work up. The authors described the advantages of their method over the current industrial process. They proposed the mechanisms of the reactions and gave the confirmation. I believe that the manuscript should be accepted after minor revisions.

1) I would recommend you to estimate influence of particle size of sodium. Try to conduct the experiment with little pieces of sodium instead of dispersed sodium. Even if the reaction is not working it would help a lot for other chemists to save time to evaluate this possibility.

Response: thanks very much for the question, more fine sodium is hard for us to prepare, but we have tried our best to cut the sodium lump to very tiny pieces, however, the reaction speed is still slower than using SD. This was expected and could be rationalized.

2) In SI, please, point out the conditions for HPLC isolations. There is no data about solvent, flow rate, retention times, etc.

Response: thanks very much for the suggestion. The HPLC we used was a recycling preparative HPLC (Gel permeation chromatography) with two columns (20 mm I.D. -600 mm L; JAIGEL-1H and JAIGEL-2H). All the products were purified by the same

conditions: CHCl_3 as eluent (100 %), flow rate: 4.0 mL/min, and the spectra was recorded at a moving paper in the speed of 60 mm/h.

Full view of HPLC

Type of HPLC

Eluting conditions

Recording device

Fig. R3 Photos of HPLC device

3) In SI, please, point out the solvent or system of solvents and ratio for every product which you isolated using recrystallization.

Response: The solvent or system of solvents and ratio for recrystallization has been added in the SI.

4) Please recheck manuscript and SI for the typos. E.g. in SI you should write Na_2SO_4 instead of NaSO_4 .

Response: Thank you for your careful examination. We have rechecked the whole manuscript and SI carefully and corrected some mistakes.

5) In SI p. S2 (2.1 General method A) I would recommend you to write solvent instead of particular THF or 1,4-dioxane in general procedure (p. S2).

Response: Thank you very much. It has been corrected in the general method A in SI.

6) In SI p. S3 in (1) For the synthesis of $\text{Ph}_2\text{P}(\text{O})\text{H}$ you should clarify and write, for example, According to General method A, sodium diphenylphosphinite 2 was obtained in 1,4-dioxane and then to describe preparation of $\text{Ph}_2\text{P}(\text{O})\text{H}$.

Response: We have corrected the content in SI. page S3.

7) In SI p. S3 synthesis of $\text{Ph}_2\text{P}(\text{O})\text{C}(\text{O})\text{Mes}$. It is important product for industry which you isolate using HPLC. Please, try to isolate it using recrystallization if it is possible.

Response: thanks very much for the suggestion. Unfortunately, we found that recrystallization of $\text{Ph}_2\text{P}(\text{O})\text{C}(\text{O})\text{Mes}$ is hard, but we could isolate and purify it by column chromatography (Hexane/EtOAc = 4/1 as eluent), which has been rewritten in the SI.

8) In General information you wrote that scale up synthesis of products were purified by column chromatography, but in the protocol (p. S6) you wrote that recrystallization was used to purify the product. What method of purification was really used?

Response: We corrected the procedure. The crude $\text{Ph}_2\text{P}(\text{O})\text{H}$ products were first filtrated by column chromatography with hexane to remove the oil from SD, then with EtOAc to wash out the product. Recrystallization was conducted using CHCl_3 and hexane. We have changed the corresponding content in the manuscript.

9) Please, measure melting point for all solid compounds and compare it with the literature data.

Response: Melting point for all solid compounds have been measured and added in SI.

10) In SI p. S12, Product 2-1, what signals are in the upfield ^1H NMR?

Response: The signals in the upfield are perhaps the peaks of trace of oil from SD and EtOAc. We washed and purified the product again and new ^1H , ^{13}C , ^{31}P NMR spectra of $\text{Ph}_2\text{P}(\text{O})\text{H}$ have been attached in SI, new characterization has also been rewritten.

11) In SI you wrote “This compound is known” and give the reference. I would recommend you to write that your data “are in agreement with the literature data”, because the fact that the substance is known doesn’t show that your data correspond to the literature.

Response: Thank you for your good suggestion. Proper expression sentences have been written in SI.

12) Please, provide the data about multiplicity and spin-spin coupling constants in ^1H NMR, although in ^1H NMR of **3'a** (p. S18) and **3'b** (p. S19), for example, I can see triplets in downfield. **3'a** 7.56 (t, 2H, $J = ?$), 7.42 (t, 2H, $J = ?$). **3'b** 7.5 (t, 2H, $J = ?$), 7.34 (t, 2H, $J = ?$).
 ^1H NMR of **3'a** and **3'b** the reference signal is 7.256 and 7.202 respectively. Please, correct the reference and other signals in these spectra.

Response:

Thanks very much for your indication to the mistake. Actually, magnify the spectra, we can see only one group of triplets in **3'a** and **3'b**, respectively.

The original expression of triplets in ^1H NMR of **3'a**, 7.59–7.56 (m, 2H) have been corrected to 7.57 (t, 2H, $J = 7.6$ Hz);

Reference signal of **3'b** has been corrected to 7.256 ppm, and the original expression of triplets in ^1H NMR of **3'b**, 7.51–7.48 (m, 2H) has been corrected to 7.55 (t, 2H, $J = 7.2$ Hz).

Reviewer #3 (Remarks to the Author):

The revised manuscript is devoted to the well-known problem of how to recycle triphenylphosphine oxide back into organic synthesis. This compound is a waste byproduct of many transformations including triphenylphosphine. The importance of the latter is overwhelming as it's widely used both in organic synthesis (production of fine chemicals) and transition metal catalysis (the synthesis of otherwise difficultly accessible organic molecules). Presented attempts in recycling triphenylphosphine oxide are of highly practical value as it gives an access to very valuable reactants in a very facile and efficient way using cheap and readily accessible reagent. Regarding this I would recommend the editorial office to accept the manuscript for publication after minor style and language improvement.

Response:

Thank you very much for your approval on our research. We have carefully modified the language and also corrected some errors in the manuscript.

Reviewers' comments:

Reviewer #1 (Remarks to the Author):

The referees' concerns were well reflected in this revised manuscript. The manuscript can be accepted provided that another minor revision is given in the following point:
1) Formation of 3 with PhNa in Fig 7 and Fig 8 is complementary to the reaction with PhLi reported by Furukawa (in ref 28). The present case kicked out PhNa, while Furukawa's reaction with PhLi retained P-Ph bond in the product. This difference should be highlighted perhaps by adding comparison scheme putting one actual example by Furukawa. Discussion to address why use of Na counter cation results in elimination of PhNa should be given. It can be scientifically interesting comparison.

Reviewer #2 (Remarks to the Author):

I believe that the revised manuscript can be published as is.

Responds to reviewers' comments:

Reviewer #1 (Remarks to the Author):

The referees' concerns were well reflected in this revised manuscript. The manuscript can be accepted provided that another minor revision is given in the following point:
1) Formation of 3 with PhNa in Fig 7 and Fig 8 is complementary to the reaction with PhLi reported by Furukawa (in ref 28). The present case kicked out PhNa, while Furukawa's reaction with PhLi retained P-Ph bond in the product. This difference should be highlighted perhaps by adding comparison scheme putting one actual example by Furukawa. Discussion to address why use of Na counter cation results in elimination of PhNa should be given. It can be scientifically interesting comparison.

Respond: Thanks for your good suggestion. The difference between the reactions of $\text{Ph}_3\text{P}(\text{O})$ with PhNa and PhLi has been illustrated in Fig. 7 and a discussion has been added in the manuscript.